# The Complex Interplay Between Dental Anxiety, Generalized Anxiety, and Dental Neglect and Oral Health Quality of Life in the General Public

**DOI:** 10.3390/healthcare13121382

**Published:** 2025-06-10

**Authors:** Abdullah S. Bin Rahmah, Mohammed I. Alsaif, Abdallah Y. Naser

**Affiliations:** 1Department of Periodontics and Community Dentistry, College of Dentistry, King Saud University, Riyadh 11362, Saudi Arabia; arahmah@ksu.edu.sa; 2Department of Applied Pharmaceutical Sciences and Clinical Pharmacy, Faculty of Pharmacy, Isra University, Amman 11622, Jordan; abdallah.naser@iu.edu.jo

**Keywords:** anxiety, dental anxiety, generalized anxiety, neglect, oral health, quality of life

## Abstract

**Background:** Dental anxiety and generalized anxiety are common psychological conditions and can lead to poor oral health and avoidance behavior. This research aims to study the complex interplay between dental anxiety, generalized anxiety, dental neglect, and oral health-related quality of life (OHRQoL) in the general public. **Methods:** This is an online survey study that was conducted between August to September 2024 in Saudi Arabia and Jordan. Four previously validated instruments were used in this study. This study made use of the Modified Dental Anxiety Scale, the General Anxiety Disorder-7, dental neglect scale, and the Oral Health Impact Profile-14. Mediation analysis was conducted using the PROCESS macro (Model 4) to explore whether dental anxiety and dental neglect mediates the relationship between generalized anxiety and OHRQoL. **Results:** This study had 2704 participants in total. Higher anxiety levels were associated with slightly lower dental neglect and significantly poorer OHRQoL. Generalized anxiety significantly predicted dental anxiety, with a coefficient of *b* = 0.275 (*p* < 0.001). Both generalized anxiety and dental anxiety were significant predictors of OHRQoL, with generalized anxiety showing stronger effect. Dental anxiety partially mediates the impact of generalized anxiety on oral health-related quality of life, while a strong direct effect remains. **Conclusions:** Dental neglect served as a minor mediator, and the primary relationship between anxiety and oral health-related quality of life is direct. Moreover, dental anxiety partially mediates the impact of generalized anxiety on oral health-related quality of life, while a strong direct effect remains.

## 1. Introduction

Dental anxiety is a psychological condition characterized as apprehension or fear associated with dental clinics or dental procedures, which may be manifested by prior pain experiences, self-reported poor oral health, avoidance behavior, and severe phobic anxiety [1,2]. Globally, dental anxiety is widespread and influences a significant proportion of the population [3,4,5,6]. However, dental anxiety occurrence is affected by many aspects, including social, cultural, economic, and environmental situations, as well as individual factors, such as prior dental experiences, gender, and age [7,8,9,10,11,12,13,14,15]. Higher education level, female gender, and lower socioeconomic status were identified as influencing factors that are associated with dental anxiety [16]. Previous literature reported that adults show higher level of dental anxiety, particularly for surgical procedures, tooth extraction, local anesthetic injection, and drilling [12,17]. Therefore, these factors may have contributed to the wide variation in the prevalence of dental anxiety from approximately 4% to more than 50%. A meta-analysis by Silveira et al. reported that the global estimated prevalence of dental fear and anxiety (DFA), high DFA, and severe DFA in adults were 15.3% (95% CI 10.2–21.2), 12.4% (95% CI 9.5–15.6), and 3.3% (95% CI 0.9–7.1), respectively [18].

Dental neglect, a continuous failure to take essential precautions to protect oral health, such as preventing infection and pain, is reported to be more likely to be present among individuals who suffer from dental anxiety [19,20]. Both problems significantly affect oral health [21,22,23]. Prior studies have declared that avoiding dental care can result in considerable health concerns, including decreased oral health-related quality of life (OHRQoL), increased toothache episodes, and increased prevalence of extracted and decayed teeth [24,25,26]. Avoiding dental care also significantly influences an individual’s social relationships, academic or occupational performance, and daily routine [27]. Moreover, research indicates that individuals at increased risk of dental problems are more likely to have poorer mental health [28]. For instance, an earlier systematic review concluded that there was a strong positive association between dental anxiety level and the level of other psychiatric disorders, mood disorders, and comorbid phobias [29]. Likewise, a prior cross-sectional study conducted in Croatia found a significant association between avoidance behavior and increased anxiety [17]. Consequently, oral health, general health, and quality of life (QoL) will increasingly deteriorate among individuals with dental anxiety because of associated poor mental health.

Generalized anxiety could have an indirect effect on dental anxiety and dental neglect, which ultimately might affect OHRQoL. Therefore, higher generalized anxiety could lead to a higher level of dental anxiety and thus being more prone to dental neglect. Dental neglect behavior could lead to poorer OHRQoL. Like many other countries, many patients suffer from dental anxiety in Jordan and Saudi Arabia [30,31,32]. Since dental anxiety influences various health aspects based on factors like personal experiences and cultural background, there is a necessity to understand the complex interaction of dental anxiety in these regions where dental anxiety is highly prevalent. There is no research on the mediating effect of dental anxiety and dental neglect on the association between generalized anxiety and OHRQoL. Therefore, this research aims to study the complex interplay between dental anxiety, generalized anxiety, dental neglect, and OHRQoL in the general public. This study hypothesizes that dental anxiety and dental neglect mediate the relationship between generalized anxiety and OHRQoL.

## 2. Methods

### 2.1. Study Design and Population

This is an online survey study that was conducted between August to September 2024 in Saudi Arabia and Jordan. Individuals from the general public served as the study population for this investigation. Participants who were 18 years of age or older and currently reside in one of the two nations met the inclusion requirements.

### 2.2. Inclusion and Exclusion Criteria

The inclusion criteria were adults aged above 18 and currently living in Jordan or Saudi Arabia. The exclusion criteria were participants aged below 18, currently living outside these countries, or did not provide consent to participate in this study.

### 2.3. Sampling Technique

The targeted study population were invited to participate in this study through social media platforms (WhatsApp, Facebook, and X). This study utilized the convenience sampling technique to recruit the study participants and invite them to participate. Anyone could access the questionnaire link. The study inclusion criteria were mentioned in the cover letter of the study.

### 2.4. Data Collection Instruments

Four previously validated instruments were used in this study. This study made use of the Modified Dental Anxiety Scale (MDAS), the General Anxiety Disorder (GAD)-7, dental neglect scale, and the Oral Health Impact Profile (OHIP)-14 [33,34,35,36,37]. The GAD-7 scales use a 4-point Likert scale, with “nearly every day” receiving a score of four and “not at all” receiving a score of zero. The GAD-7 scale has a maximum score of 21. Mild anxiety is indicated by a score between 5 and 9, moderate anxiety by a score between 10 and 14, and severe anxiety by a score between 15 and 21. The seven dimensions of the influence that oral health issues have on people’s quality of life—functional limitation, physician pain, psychological discomfort, physical disability, psychological disability, social impairment, and handicap—are examined by this questionnaire tool. A 5-point Likert scale, with never (scoring zero) and very often (scoring four) as the extremes, is used to score the 14 questions. Fifty-six is the highest possible OHIP-14 score. The more severe the oral health issues, the greater the influence of oral health on people’s quality of life. The dental neglect scale is comprised of 6-items that are assessed using a 5-point scale, with answers ranging from “Definitely no” to “Definitely yes”. Possible scores range from 6 to 30, with higher scores indicative of greater dental neglect.

### 2.5. Sample Size Calculation

The required sample size from each study population was 385 participants, based on a confidence interval of 95%, a standard deviation of 0.5, and a margin of error of 5%.

### 2.6. Ethical Approval

The research ethics committee at Isra University, Amman, Jordan, approved the study protocol (SREC/24/09/114, Date: August 2024). Informed consent was obtained from the study participants prior to study commencement. This study was conducted in accordance with the World Medical Association (WMA) Declaration of Helsinki.

### 2.7. Response Rate and Handling of Incomplete Responses

This study utilized an online survey, and the questionnaire link was distributed using convenience sampling technique through social media platforms. As such, we were not able to estimate the response rate for this study as we do not have an estimate for the number of participants who received the questionnaire link. In this study, we did not have any incomplete data as the online survey asked the participants to answer all questions before submission.

### 2.8. Statistical Analysis

Categorical variables were presented using frequencies and percentages. Continuous variables were presented as mean and standard deviation. Chi-squared test was used to examine the difference in the proportion between different categorical variables. Student *t*-test was used to examine the difference in continuous variables. A mediation analysis was conducted using the PROCESS macro (Model 4) in SPSS version 29 to explore whether dental anxiety and dental neglect mediates the relationship between generalized anxiety and OHRQoL. GAD-7 score was entered as the independent variable (X), MDAS score or dental neglect score as the mediators (M), and OHIP-14 score as the dependent variable (Y) (Figure 1). Bootstrapping with 5000 samples was used to estimate the indirect effects, and a 95% confidence interval (CI) was calculated. The analysis provided the total effect, direct effect, and indirect effect of generalized anxiety on OHRQoL, controlling for the mediating effect of dental neglect. Results include standardized coefficients and significance levels, and interactions between generalized anxiety and dental neglect were also tested. Mplus software version 8.1 was used to perform structural equation modelling. Estimation of both direct and indirect effects for each pathway was facilitated through the use of SEM within the Mplus software.

## 3. Results

This study had 2704 participants in total. Approximately 53.9% of them were female; 28.6% of the study sample was composed of people in the age range of 31 to 40. Eighty-one percent of the participants were unmarried. Approximately 55.0% of those who took part said they had a bachelor’s degree. Approximately one-third (38.5%) of the study participants stated that their household makes less than USD 700 per month. Of the participants, about 41.1% said they were unemployed. Smoking was reported by nearly one-fifth (18.5%) of the study participants. Approximately 10% of research participants (9.9%) stated that they had a history of comorbidities. For further details on the demographic characteristics of the study participants, refer to Table 1.

### 3.1. Anxiety Status Among the Study Sample

#### 3.1.1. Generalized Anxiety Status

The mean GAD-7 score for the study participants was 9.1 (SD: 5.6) out of 21. The mean GAD-7 score in Jordan and Saudi Arabia was 10.5 (SD: 5.8) and 6.8 (SD: 5.5), respectively (*p* < 0.001). The prevalence of severe anxiety (a total score of 15 and above) among study sample was 21.4%. Table 2 below presents anxiety status of the study sample.

#### 3.1.2. Dental Anxiety Status

The mean dental anxiety score among the study participants was 12.4 (SD: 5.0) out of 25, which indicates a moderate level of dental anxiety among the study participants. The mean dental anxiety score in Jordan and Saudi Arabia was 12.5 (SD: 5.2) and 12.3 (SD: 4.7), respectively (*p* = 0.313). Around 39.5% of the study sample showed a moderate level of dental anxiety. A total of 13.8% of the study sample showed a high level of dental anxiety.

#### 3.1.3. Dental Neglect Profile

The mean dental neglect score for the study participants was 20.5 (SD: 4.3) out of 30, which is equal to 68.3%. The mean dental neglect score in Jordan and Saudi Arabia was 20.5 (SD: 4.3) and 20.6 (SD: 4.2), respectively (*p* = 0.251). This dental neglect score highlights that the study participants have moderate to high levels of dental neglect.

#### 3.1.4. First Moderator (Dental Neglect)

Dental neglect served as a mediator to investigate the relationship of generalized anxiety with OHRQoL (Figure 1). The results have shown that generalized anxiety has had a small but statistically significant negative direct effect on dental neglect (b = −0.054, *p* < 0.001), which shows that the higher the anxiety, the lower the dental neglect. The effect of generalized anxiety on OHRQoL is still directly significant at b = 0.855 with *p* < 0.001. The total effect of generalized anxiety on OHRQoL, on the other hand, has shown to be b = 0.876 with *p* < 0.001. The indirect effect via dental neglect is statistically significant, albeit small, with b = 0.021, 95% CI (0.010, 0.034), and this supports the notion that the main part of the relationship between anxiety and OHRQoL is determined via the direct effect of anxiety although dental neglect functions as a minor mediator.

#### 3.1.5. Second Moderator (Dental Anxiety)

The mediation analysis tested whether dental neglect mediates the relationship between generalized anxiety and OHRQoL, Figure 1. It showed that generalized anxiety significantly predicted dental neglect with a b of 0.275 and a *p*-value of less than 0.001, indicating that the higher the generalized anxiety, the higher the dental anxiety. Overall, generalized anxiety accounted for 10.6% of the variance in dental neglect, as indicated by an R^2^ value of 0.106. In predicting OHRQoL, generalized anxiety and dental neglect were significant predictors. Generalized anxiety was found to have a direct positive effect on OHRQoL, with a coefficient of b = 0.757 (*p* < 0.001), while the contribution of dental neglect to OHRQoL was also significant (b = 0.432; *p* < 0.001). The models explain 20.0% of the total variance in OHRQoL, with R^2^ = 0.200. The total effect of generalized anxiety on OHRQoL cumulatively was significant (b = 0.876, *p* < 0.001), but the indirect effect from dental neglect was significant too, b = 0.119, 95% CI (0.090, 0.149), which meant partial mediation. The standardized indirect effect was 0.056. This means that dental neglect partially mediates the relationship between generalized anxiety and OHRQoL, while there is still a high influence of generalized anxiety on OHRQoL.

#### 3.1.6. Predictors of Dental Anxiety and Generalized Anxiety

Binary logistic regression analysis identified that males and higher income level (above USD 1500) were predictors of lower likelihood of experiencing severe dental anxiety level (*p* < 0.05). Females, unemployed, university students, smokers, and those with comorbidities history had higher likelihood of experiencing severe generalized anxiety level (*p* < 0.05), Table 3.

### 3.2. Structural Equation Modelling

In the pathway from general anxiety to OHRQoL, the sum of the indirect effects of dental anxiety and dental neglect was significant (β = 0.14, SE: 0.01, *p*-value = <0.001). The indirect effects of dental anxiety and neglect were also substantial, Figure 2, Table 4.

## 4. Discussion

Surprisingly, our study found that individuals with higher anxiety levels showed lower levels of dental neglect. Multiple previous studies have highlighted that the use of or access to dental health services is low among patients with psychiatric conditions, particularly those with anxiety disorders [38,39]. A prior study among adult patients with dental anxiety in Sweden reported that, compared to regular dental attendees, anxiety (general and dental) among those who regularly attended dental clinics was lower [40]. Furthermore, regardless of the severity of oral disease, anxiety may increase the perception of dental pain, as in the case of burning mouth syndrome, which is often associated with anxiety [41]. In addition, patients with anxiety disorders do not receive treatment until the pain becomes severe, which increases the risk of dental neglect [42,43]. Earlier investigations identified several barriers to adherence to treatment and attendance at general dental clinics among patients with dental anxiety, including cancelling appointments due to illness after visiting the clinic and other barriers related to transportation and scheduling appointments by phone [44,45].

On the other hand, a prior study conducted in the United States declared that after adjusting for body mass index and other health behaviors among pregnant women, anxiety disorders did not show a significant association with tooth loss and dental visits related to oral health consequences [46]. Still, in line with our findings, higher levels of depression are associated with increasing oral hygiene among dental students in Iraq [47], which indicates lower dental neglect levels. Patients with depression feel that they must adopt a more proactive approach to fulfill the anticipations of others [48], which may also result in decreased dental neglect levels among them. The differences in the findings between our study and previous studies could be attributed to the differences in study samples and cultural contexts. In Eastern countries such as Saudi Arabia and Jordan, people adopt different healthcare behaviors from Western countries. This highlights the importance of developing interventions that target generalized and dental-anxiety that take into consideration cultural variations.

The variety in the presence or absence of associations, positive or negative, between anxiety with dental neglect across different research could be attributed to several aspects, such as individual, cultural, economic, and educational factors. A prior study documented that dental neglect was significantly influenced (*p* < 0.05) by education level, income, and age among Hong Kong populations [49]. Most participants in several previous investigations conducted in Nigeria, India, and Pakistan had limited educational levels and low socioeconomic status, and they avoided attendance at the dental clinic until they had severe swelling and pain [50,51,52]. These suggest a possible link between low socioeconomic and education levels and dental neglect. In our study, most participants have sufficient educational levels (bachelor’s degree: 55.0% and higher education: 4.8%) and good socioeconomic status, which may decrease dental neglect in patients with higher levels of anxiety. Still, research from Jordan and Pakistan found that busy schedules among participants with higher than matriculation education acted as a barrier to the use of dental services [52,53]. Moreover, time constraints and busy schedules were also reported as barriers to using dental services among adults from Saudi Arabia [54]. Accordingly, dental and mental health experts may benefit from training skills and awareness, along with customized interventions, to treat psychiatric comorbidity patients who experience dental health issues [55,56].

Although our study found that higher levels of anxiety were associated with decreased dental neglect, anxiety level and dental neglect affected an individual’s OHRQoL. Consistent with the present results, previous studies have demonstrated a significant association between low OHRQoL and high dental neglect [49,57]. These associations could be attributed to multiple factors; the most common of these factors that act as barriers to dental care use is dental anxiety, as mentioned by prior research [58]. Hence, dental-related neglect, avoidance, and anxiety should be managed to improve and promote the interventional approach [59], subsequently improving OHRQoL and overall health and QoL. A previous study by Supriya et al. found correlation between dental fear and oral health behavior, which suggests that dental anxiety may result in adopting more proactive oral hygiene practices [60].

Aligning with our observations, an earlier cohort study reported that despite dental anxiety, anxiety symptoms impact OHRQoL among young women in Brazil [61]. In Norway, there was a demonstrated strong association between general anxiety and OHRQoL; higher anxiety was associated with worse OHRQoL [62]. Additionally, probable anxiety was associated with worse OHRQoL among adults in Germany [63]. A recent meta-analysis and systematic review concluded that there were significant indications that anxiety was among the potential aspects that impact OHRQoL [64]. In line with this, several previous studies declared that anxiety was significantly associated with worse OHRQoL [65,66,67]. This strengthens the evidence that there is a connection between anxiety and OHRQoL, as demonstrated across participants from different countries and settings. Higher anxiety and depression scores were associated with worse OHRQoL, as reported by earlier research conducted in Michigan, Korea [68], and Romania [43,69]. Several factors could explain this observation. For example, anxiety was linked with numerous adverse consequences on individuals and societies, including increased healthcare expenditures [70,71,72]; when healthcare expenses increase, patients, especially those with low incomes, tend to avoid attendance at dental clinics and consequently, OHRQoL for these individuals will be worse. Another possible explanation for these findings is that psychiatric conditions, including anxiety, cause behavioral and psychological changes [73,74,75], reduce social interaction, impact routine activities, and restrict individuals’ freedom [73,76]. This demonstrates the importance of psychosocial aspects as they influence important health behaviors such as oral hygiene habits. Consistent with this explanation, the psychological discomfort domain, along with the physical pain domain, has been shown to have higher OHIP scores in Saudi Arabia [77,78]. Hence, managing dental neglect and anxiety is essential to enhance OHRQoL. Interventions aimed at promoting awareness of oral health could be helpful [79].

Further crucial findings from the current investigation are that the higher generalized anxiety level was associated with higher dental anxiety. Also, a higher level of generalized anxiety had a direct positive effect on OHIP-14. Dental anxiety affects individuals’ OHRQoL. Our results match those observed in the literature. For instance, in Saudi Arabia [80], Turkey [81], and Canada [82], earlier investigations reported positive associations between generalized anxiety and dental anxiety. A prior study conducted in Norway showed that a higher level of generalized anxiety was associated with worse OHRQoL [62]. Moreover, considerable previous research from diverse regions highlighted that dental anxiety influences individuals’ OHRQoL, as they demonstrated negative associations between dental anxiety and OHRQoL [45,83,84,85,86,87,88,89,90,91,92]. These highlighted the high prevalence of these issues, which underlines that interventions aimed at decreasing generalized anxiety and dental anxiety are required to improve OHRQoL. Targeted psychosocial interventions such as cognitive-behavioral therapy are recommended in order to decrease anxiety and ultimately enhance OHRQoL [93,94]. Finally, it is essential to consider factors related to mental disorders when designing dental care plans and interventions, which may aid in enhancing OHRQoL.

This study has multiple strengths. This study examined the relationship between dental anxiety, generalized anxiety, and dental neglect and oral health quality of life in the general public using two integrated approaches (mediation analysis and structural equation modelling), which model the relationships across the study variables and assess the overall model fit through the use of mediation analysis. At the same time, structural equation modelling estimates all path coefficients concurrently, providing a more comprehensive understanding of the direct and indirect effects within the model. This offers an advantage over previous studies that examined the associations and correlations independently. Moreover, this study provided insights from two Middle Eastern countries, which are still underrepresented in the literature in this area of oral health research. This study has limitations. The use of an online cross-sectional survey study restricted the ability to examine causality across the study variables and has limited generalizability to the targeted study population. Moreover, self-administered questionnaires are prone to social desirability bias and reporting bias. Despite that, this study has adequate statistical power, supported by the large sample size from the two countries. Therefore, our study findings should be interpreted carefully. Future longitudinal studies are warranted to examine causality. These studies should involve clinical assessment to validate self-reported data and have more accurate and objective measures presenting oral health. Moreover, future research should develop interventions that support both dental and mental health of high-risk populations.

## 5. Conclusions

Dental neglect served as a minor mediator, and the primary relationship between anxiety and oral health-related quality of life is direct. Moreover, dental anxiety partially mediates the impact of generalized anxiety on oral health-related quality of life, while a strong direct effect remains. Healthcare professionals are encouraged to screen for and manage generalized and dental anxiety and give them more priority due to their direct impact on OHRQoL. This could be achieved through regular screening and early interventions to manage them. Mental health awareness campaigns are warranted for both patients and healthcare professionals. This strengthens the ability of healthcare professionals to manage the anxiety of their patients.

## Figures and Tables

**Figure 1 healthcare-13-01382-f001:**
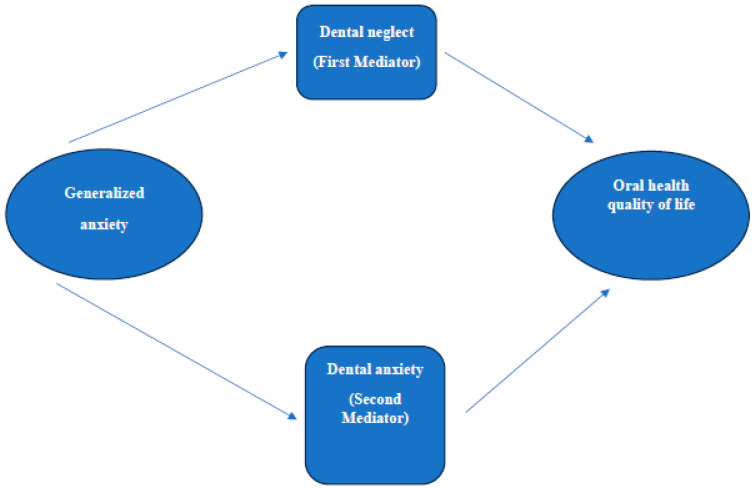
The mediating effect of dental anxiety and dental neglect on the association between generalized anxiety and oral health quality of life.

**Figure 2 healthcare-13-01382-f002:**
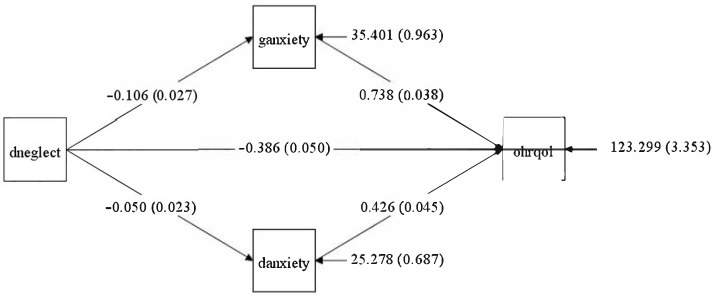
SEM path diagram of the association between general anxiety and Oral Health-related Quality of Life mediated by dental anxiety and dental neglect.

**Table 1 healthcare-13-01382-t001:** Participants’ demographic characteristics.

Variable	Overall		Jordan (*n* = 1684)		Saudi Arabia (*n* = 1020)		*p*-Value
	Frequency	Percentage	Frequency	Percentage	Frequency	Percentage	
**Gender**							
Females	1458	53.9%	762	45.2%	696	68.2%	<0.001
Males	1246	46.1%	922	54.8%	324	31.8%	
**Age**							
18–23 years	587	21.7%	278	16.5%	309	30.3%	<0.001
24–30 years	678	25.1%	467	27.7%	211	20.7%	
31–40 years	772	28.6%	355	21.1%	417	40.9%	
41–50 years	278	10.3%	117	6.9%	161	15.8%	
51–60 years	219	8.1%	150	8.9%	69	6.8%	
61 years and older	170	6.3%	89	5.3%	81	7.9%	
**Marital status**							
Single	2197	81.3%	1554	92.3%	643	63.0%	<0.001
Married	440	16.3%	120	7.1%	320	31.4%	
Divorced	51	1.9%	9	0.5%	42	4.1%	
Widowed	16	0.6%	1	0.1%	15	1.5%	
**Education level**							
Diploma or lower	1086	40.2%	720	42.8%	366	35.9%	<0.001
Bachelor degree	1487	55.0%	897	53.3%	590	57.8%	
Higher education	131	4.8%	67	4.0%	64	6.3%	
**Family monthly income level**							
Less than USD 700	1040	38.5%	895	53.1%	145	14.2%	<0.001
USD 700–1500	680	25.1%	522	31.0%	158	15.5%	
USD 1500–2000	408	15.1%	146	8.7%	262	25.7%	
More than USD 2000	576	21.3%	121	7.2%	455	44.6%	
**Employment status**							
Retired	73	2.7%	24	1.4%	49	4.8%	<0.001
Unemployed	1112	41.1%	744	44.2%	368	36.1%	
Working in healthcare sector	166	6.1%	29	1.7%	137	13.4%	
University student	970	35.9%	773	45.9%	197	19.3%	
Working in other sectors	383	14.2%	114	6.8%	269	26.4%	
**Smoking status**							
Smoker	500	18.5%	318	18.9%	182	17.8%	0.507
**Comorbidities history**							
Yes	268	9.9%	156	9.3%	112	11.0%	0.163

**Table 2 healthcare-13-01382-t002:** Generalized anxiety status of the study sample.

	Overall		Jordan		Saudi Arabia		*p*-Value
Variable	Frequency	Percentage	Frequency	Percentage	Frequency	Percentage	
GAD-7 Diagnosis Based on Score Distribution							
Minimal anxiety	713	26.4%	288	17.1%	425	41.7%	<0.001
Mild anxiety	757	28.0%	516	30.6%	241	23.6%	
Moderate anxiety	654	24.2%	400	23.8%	254	24.9%	
Severe anxiety	580	21.4%	480	28.5%	100	9.8%	

**Table 3 healthcare-13-01382-t003:** Predictors of dental anxiety and generalized anxiety.

Variable	Odds Ratio of Severe Dental Anxiety (95% Confidence Interval)	*p*-Value	Odds Ratio of Severed Generalized Anxiety (95% Confidence Interval)	*p*-Value
**Gender**				
Females (Reference category)	1.00		1.00	
Males	0.51 (0.39–0.67)	<0.001	0.58 (0.46–0.71)	<0.001
**Age**				
18–23 years (Reference category)	1.00		1.00	
24–30 years	0.92 (0.69–1.23)	0.575	0.51 (0.39–0.67)	<0.001
31–40 years	0.80 (0.48–1.33)	0.396	0.39 (0.23–0.64)	<0.001
41–50 years	0.39 (0.14–1.09)	0.072	0.31 (0.13–0.72)	0.007 **
51–60 years	0.50 (0.12–2.12)	0.345	0.26 (0.06–1.08)	0.064
61 years and older	1.00 (0.12–8.30)	0.997	-	
**Marital status**				
Single (Reference category)	1.00		1.00	
Married	0.77 (0.56–1.06)	0.105	0.43 (0.31–0.58)	<0.001
Divorced	0.95 (0.43–2.14)	0.909	0.79 (0.39–1.59)	0.514
Widowed	0.40 (0.05–3.04)	0.376	0.46 (0.11–2.05)	0.311
**Education level**				
Diploma or lower (Reference category)	1.00		1.00	
Bachelor degree	0.96 (0.76–1.20)	0.714	0.58 (0.48–0.70)	<0.001
Higher education	0.91 (0.53–1.56)	0.729	0.82 (0.53–1.25)	0.351
**Family monthly income level**				
Less than USD 700 (Reference category)	1.00		1.00	
USD 700–1500	0.77 (0.58–1.02)	0.065	0.68 (0.55–0.86)	<0.001
USD 1500–2000	0.59 (0.41–0.84)	0.004 **	0.45 (0.34–0.61)	<0.001
More than USD 2000	0.72 (0.53–0.97)	0.030 *	0.35 (0.26–0.46)	<0.001
**Employment status**				
Retired (Reference category)	1.00		1.00	
Unemployed	1.89 (0.81–4.43)	0.143	2.54 (1.25–5.18)	0.010 *
Working in healthcare sector	1.36 (0.52–3.58)	0.535	0.71 (0.29–1.70)	0.437
University student	2.08 (0.88–4.87)	0.093	2.05 (1.00–4.19)	0.049 *
Working in other sectors	1.12 (0.45–2.78)	0.801	0.97 (0.45–2.08)	0.939
**Smoking status**				
No (Reference category)	1.00		1.00	
Yes	0.94 (0.71–1.26)	0.689	1.26 (1.01–1.59)	0.044 *
**Comorbidities history**				
No (Reference category)	1.00		1.00	
Yes	1.04 (0.72–1.49)	0.833	1.55 (1.17–2.05)	0.002 **

* *p* < 0.05; ** *p* < 0.01.

**Table 4 healthcare-13-01382-t004:** SEM output of the association between general anxiety and Oral Health-related Quality of Life mediated by dental anxiety and dental neglect.

	β (SE)	*p*-Value
**Direct effects**		
General Anxiety → Dental Anxiety	0.28 (0.01)	<0.001
General Anxiety → Dental Neglect	−0.05 (0.01)	<0.001
General Anxiety → OHRQoL	0.74 (0.04)	<0.001
Dental Anxiety → OHRQoL	0.43 (0.04)	<0.001
Dental Neglect → OHRQoL	−0.39 (0.05)	<0.001
**Intercepts**		
Dental Anxiety	9.9 (0.17)	<0.001
Dental Neglect	21.0 (0.15)	<0.001
OHRQoL	10.85 (1.21)	<0.001
**Indirect effects**		
General Anxiety → OHRQoL		
Sum of indirect effect	0.14 (0.01)	<0.001
General Anxiety → Dental Anxiety → OHRQoL	0.12(0.01)	<0.001
General Anxiety → Dental Neglect → OHRQoL	0.02 (0.006)	<0.001
**Measuring Model Fit**		
CFI ^1^	1	
TLI ^2^	1	
RMSEA ^3^	0.947	
SRMR ^4^	0.005	

^1^ Comparative Fit Index; ^2^ Tucker-Lewis Index; ^3^ Root Mean Square Error of Approximation; ^4^ Standardised Root Mean Square Residual.

## Data Availability

The datasets supporting the conclusions of this study are available from the corresponding author upon request.

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
