# Peer review of "The Complex Interplay Between Dental Anxiety, Generalized Anxiety, and Dental Neglect and Oral Health Quality of Life in the General Public"

_healthcare, 2025, doi:10.3390/healthcare13121382_

Round 1

Reviewer 1 Report

Comments and Suggestions for Authors

“The Complex Interplay Between Dental Anxiety, Generalized Anxiety, and Dental Neglect and Oral Health Quality of Life in the General Public” is an interesting exploration of the interrelationships of anxiety and dental neglect on oral health. It addresses an important public health problem and uses a substantial convenience sample size (N=2,704), providing an appropriate sample testing the authors’ mediational hypotheses. Exploring mediation effects, moving beyond simple associations, is a particular strength of this paper.

To further strengthen the manuscript, I have a few suggestions for the authors to consider:

  1. Analyses: Although the mediation analysis using the PROCESS macro is a viable approach, the hypotheses would be more rigorously tested using a structural equation model (SEM). This approach would offer a more parsimonious way to simultaneously model the relationships among the variables and assess the overall model fit. That is, SEM estimates all path coefficients concurrently, providing a more comprehensive understanding of the direct and indirect effects within the model. Furthermore, SEM provides fit indices indicating how well the proposed model represents the observed data, offering a stronger test of the hypothesized relationships than individual PROCESS macro analyses.
  2. In the discussion section, the authors occasionally equated generalized and dental anxiety with “mental illness,” which is too broad. I recommend that the authors refine this section to focus both the consideration of their own results — and the cited literature used to contextualize the discussion — specifically on anxiety disorders
  3. Although the introduction effectively sets up the problem, some of the prevalence citations were not as on point or up to date. For example, a stronger reference on the prevalence of dental fear in adults is Silveira ER, Cademartori MG, Schuch HS, Armfield JA, Demarco FF. Estimated prevalence of dental fear in adults: a systematic review and meta-analysis. J Dent. 2021;108:103632. doi:10.1016/j.jdent.2021.103632
  4. Regarding the negative relationship between dental anxiety and dental neglect. It might be worth exploring the possibility that many anxious patients may increase their dental self-care as a response to avoiding dental professionals.

Overall, this manuscript could be a useful examination of the factors influencing oral health quality of life.

Author Response

Reviewer 1:

To further strengthen the manuscript, I have a few suggestions for the authors to consider:

- Analyses: Although the mediation analysis using the PROCESS macro is a viable approach, the hypotheses would be more rigorously tested using a structural equation model (SEM). This approach would offer a more parsimonious way to simultaneously model the relationships among the variables and assess the overall model fit. That is, SEM estimates all path coefficients concurrently, providing a more comprehensive understanding of the direct and indirect effects within the model. Furthermore, SEM provides fit indices indicating how well the proposed model represents the observed data, offering a stronger test of the hypothesized relationships than individual PROCESS macro analyses.

- Thank you for this comment, we have now addressed this point and added the findings of the SEM analysis, see Table 4 and Figure 2.

- In the discussion section, the authors occasionally equated generalized and dental anxiety with “mental illness,” which is too broad. I recommend that the authors refine this section to focus both the consideration of their own results — and the cited literature used to contextualize the discussion — specifically on anxiety disorders

- Thank you for this comment, we have now addressed this point and revised the discussion as appropriate, see pages 13 and 15.

- Although the introduction effectively sets up the problem, some of the prevalence citations were not as on point or up to date. For example, a stronger reference on the prevalence of dental fear in adults is Silveira ER, Cademartori MG, Schuch HS, Armfield JA, Demarco FF. Estimated prevalence of dental fear in adults: a systematic review and meta-analysis. J Dent. 2021;108:103632. doi:10.1016/j.jdent.2021.103632

- Thank you for this comment, we have now addressed this point and added this important up to date reference on the prevalence of dental anxiety and dental fear, see lines 79-82.

- Regarding the negative relationship between dental anxiety and dental neglect. It might be worth exploring the possibility that many anxious patients may increase their dental self-care as a response to avoiding dental professionals.

- Thank you for this comment, we have now addressed this point and discussed it further, see lines 334-336.

Reviewer 2 Report

Comments and Suggestions for Authors

Thank you for the opportunity to review this cross-sectional study, which aims to explore the complex interplay between dental anxiety, generalized anxiety, dental neglect, and oral health-related quality of life (OHRQoL) in the general public. This is a timely and relevant topic in the field of public oral health. However, several revisions are necessary to enhance the scientific clarity, methodological rigor, and alignment with reporting standards.

1. Abstract – Results Section
The results presented in the abstract are overly detailed and difficult to follow. Please simplify the presentation of key findings by summarizing them more clearly and concisely, focusing on the most relevant outcomes.

2. References
References within the manuscript are not formatted according to the journal’s citation guidelines, neither in-text nor in the reference list. Please ensure all references are properly cited and formatted in accordance with the journal’s instructions for authors.

3. Introduction
The introduction should be slightly expanded to include a broader international context regarding dental anxiety. I recommend including the following recent studies as references to strengthen the background:

https://doi.org/10.3390/ijerph20126118

https://doi.org/10.3390/medicina60081303

https://doi.org/10.3390/healthcare10122352

In addition, the introduction should clearly state the hypothesis of the study at its conclusion to provide a focused framework for the research.

4. Table 1 – Statistical Comparison
Please apply a chi-square test to Table 1 to assess whether there are significant demographic differences between the two countries included in the study. This would help determine if comparisons between the populations are valid and meaningful.

5. Methodology
The methodology section requires substantial revision. Currently, it does not follow the CHERRIES guidelines (Checklist for Reporting Results of Internet E-Surveys), which are standard for online surveys. Please revise the section accordingly, ensuring to include:

Sample size calculation and justification

Detailed inclusion and exclusion criteria

Ethical approval with date and committee

Response rate and handling of incomplete responses

6. Results – Data Structuring
All questionnaire results should be presented separately for each country. Comparative analysis should be conducted to determine if there are significant differences between the two populations.
Additionally, assess how demographic factors (e.g., age, gender, education, income) influence both dental and generalized anxiety scores using appropriate statistical tests (e.g., regression analysis, ANOVA).

7. Study Power and Limitations
The manuscript currently does not mention the statistical power of the study or discuss its limitations, which are essential components of a well-rounded scientific report. Please include a section addressing both.

Author Response

Reviewer 2:

Thank you for the opportunity to review this cross-sectional study, which aims to explore the complex interplay between dental anxiety, generalized anxiety, dental neglect, and oral health-related quality of life (OHRQoL) in the general public. This is a timely and relevant topic in the field of public oral health. However, several revisions are necessary to enhance the scientific clarity, methodological rigor, and alignment with reporting standards.

  1. Abstract – Results Section
    The results presented in the abstract are overly detailed and difficult to follow. Please simplify the presentation of key findings by summarizing them more clearly and concisely, focusing on the most relevant outcomes.

- Thank you for this comment, we have now addressed this point and summarized the key findings, see lines 43-55.

  1. References
    References within the manuscript are not formatted according to the journal’s citation guidelines, neither in-text nor in the reference list. Please ensure all references are properly cited and formatted in accordance with the journal’s instructions for authors.

- Thank you for this comment, we have now addressed this point and checked that the references are meeting the journal’s guidelines. In addition, any further formatting issue can be handled by the production office.

  1. Introduction
    The introduction should be slightly expanded to include a broader international context regarding dental anxiety. I recommend including the following recent studies as references to strengthen the background:

https://doi.org/10.3390/ijerph20126118

https://doi.org/10.3390/medicina60081303

https://doi.org/10.3390/healthcare10122352

In addition, the introduction should clearly state the hypothesis of the study at its conclusion to provide a focused framework for the research.

- Thank you for this comment, we have now addressed this point in the introduction, see lines 74-78 and lines 107-108.

  1. Table 1 – Statistical Comparison
    Please apply a chi-square test to Table 1 to assess whether there are significant demographic differences between the two countries included in the study. This would help determine if comparisons between the populations are valid and meaningful.

- Thank you for this comment. The aim of this study was to gain insights from two Middle eastern countries not to make direct comparison. However, based on the reviewer’s comment, we have now conducted chi-square test and estimated the p-value, see Table 1.

  1. Methodology
    The methodology section requires substantial revision. Currently, it does not follow the CHERRIES guidelines (Checklist for Reporting Results of Internet E-Surveys), which are standard for online surveys. Please revise the section accordingly, ensuring to include:

Sample size calculation and justification

Detailed inclusion and exclusion criteria

Ethical approval with date and committee

Response rate and handling of incomplete responses

- Thank you for this comment, we have now addressed this point in the method section, see pages 4 and 5.

  1. Results – Data Structuring
    All questionnaire results should be presented separately for each country. Comparative analysis should be conducted to determine if there are significant differences between the two populations.
    Additionally, assess how demographic factors (e.g., age, gender, education, income) influence both dental and generalized anxiety scores using appropriate statistical tests (e.g., regression analysis, ANOVA).

- Thank you for this comment, we have now added these findings to the results section based on the reviewer’s comment, see lines 203-225.

  1. Study Power and Limitations
    The manuscript currently does not mention the statistical power of the study or discuss its limitations, which are essential components of a well-rounded scientific report. Please include a section addressing both.

- Thank you for this comment, we have now addressed this comment and added study limitations and commented on the statistical power of this study, see lines 381-389.

Reviewer 3 Report

Comments and Suggestions for Authors

Dear Authors,
Thank you for submitting an interesting manuscript. Here are the instructions and questions that need to be addressed in order to make your manuscript suitable for possible publication:

  1. I kindly ask the authors to revise the beginning of the introduction. Specifically, dental anxiety and dental fear (or phobia) are not synonyms; more precisely, dental anxiety is considered the “lowest form” of dental fear. Therefore, this distinction should be accurately addressed, especially since the focus of the paper is on dental anxiety.

  2. In the introduction, the authors should more precisely address dental anxiety in the context of previous research. Please clearly define the research gap that led to the formulation of the hypothesis and aims of this study.

  3. Materials and Methods – please include the following:
    a. Inclusion and exclusion criteria
    b. The exact online platform used for the distribution of the questionnaires
    c. Whether the invitation was sent out individually or if anyone could access the questionnaire
    d. Whether you obtained ethical board approval for the dissemination of the results, since this is a human study involving protected data

  4. Results:
    a. Table 1 – please include a row with data separated by gender (males are missing)
    b. Figure 1 is too large and should be presented within the context of the Materials and Methods section
    c. Sections 3.1.2 – 3.1.5 – the results should be presented in tabular or graphical form.

  5. The discussion is well written and detailed; however, please add a paragraph discussing the limitations of this study, as well as future directions and unanswered questions that emerged from it.

Wish you good luck.

Author Response

Reviewer 3:

Thank you for submitting an interesting manuscript. Here are the instructions and questions that need to be addressed in order to make your manuscript suitable for possible publication:

  1. I kindly ask the authors to revise the beginning of the introduction. Specifically, dental anxiety and dental fear (or phobia) are not synonyms; more precisely, dental anxiety is considered the “lowest form” of dental fear. Therefore, this distinction should be accurately addressed, especially since the focus of the paper is on dental anxiety.
  • Thank you for this comment, we have now addressed this point in the introduction, see line 69.
  1. In the introduction, the authors should more precisely address dental anxiety in the context of previous research. Please clearly define the research gap that led to the formulation of the hypothesis and aims of this study.
  • Thank you for this comment, we have now highlighted the findings from previous research in the introduction and highlighted the rationale for this study in addition to the study’s hypothesis, see the introduction page 4.
  1. Materials and Methods – please include the following:
    a. Inclusion and exclusion criteria
    b. The exact online platform used for the distribution of the questionnaires
    c. Whether the invitation was sent out individually or if anyone could access the questionnaire
    d. Whether you obtained ethical board approval for the dissemination of the results, since this is a human study involving protected data

Thank you for this comment, we have now addressed this point in the method section, see lines 115-168.

  1. Results:
    a. Table 1 – please include a row with data separated by gender (males are missing)
  • Thank you for this comment, we have now addressed this point in Table 1.
    Figure 1 is too large and should be presented within the context of the Materials and Methods section
  • Thank you for this comment, we have now addressed this point and presented the figure in the method section. The size of the graph will be corrected by the production team based on the journal’s guidelines.
    Sections 3.1.2 – 3.1.5 – the results should be presented in tabular or graphical form.
  • Thank you for this comment, we preferred to keep these findings narratively to avoid having large number of tables in the manuscript.
  1. The discussion is well written and detailed; however, please add a paragraph discussing the limitations of this study, as well as future directions and unanswered questions that emerged from it.

Thank you for this comment, we have now addressed this point and added the study limitations and future directions, see lines 381-389.

Reviewer 4 Report

Comments and Suggestions for Authors

Dear authors,

below are comments related to your research:

In the Introduction chapter:

In the introduction, you presented the basic relationships between the research variables, but I advise you to further explain how general anxiety, dental anxiety, dental neglect, and oral health are interconnected within a single theoretical framework. It is necessary to indicate whether general anxiety can indirectly affect oral health through dental anxiety and dental neglect. I also suggest that you include several recent scientific papers dealing with this topic or related constructs in the introduction, in order to strengthen the theoretical basis of the paper.

In the Discussion chapter:

  1. The discussion is based on a solid summary of the main findings and their comparison with previous research, but I recommend a more precise and detailed connection of the results to the existing literature. Currently, comparisons with other research are often general, which reduces the power of interpretation.

 It is necessary to include a critical interpretation of the results, especially when they differ from the findings of previous studies. Consider possible reasons for such discrepancies – e.g. differences in sample, methodology, cultural context, etc. If your research results are consistent with previous research, it is important to clearly highlight this consistency and explain its implications for the scientific community and clinical practice.

  1. The contribution of this study to existing knowledge needs to be highlighted more clearly. Authors should indicate how their research fills gaps in the scientific literature or offers new insights into the relationship between dental anxiety, general anxiety, dental neglect and oral health-related quality of life.
  2. The practical implications of this study are not sufficiently developed. It would be useful to highlight how the results can be applied in clinical practice, education or public health interventions.
  3. I recommend that the final part of the discussion include specific suggestions for future research that logically follow from the results of this study.

Author Response

Reviewer 4:

below are comments related to your research:

In the Introduction chapter:

In the introduction, you presented the basic relationships between the research variables, but I advise you to further explain how general anxiety, dental anxiety, dental neglect, and oral health are interconnected within a single theoretical framework. It is necessary to indicate whether general anxiety can indirectly affect oral health through dental anxiety and dental neglect. I also suggest that you include several recent scientific papers dealing with this topic or related constructs in the introduction, in order to strengthen the theoretical basis of the paper.

  • Thank you for this comment, we have now addressed this point and explained how general anxiety, dental anxiety, dental neglect, and oral health are interconnected within a single theoretical framework. Besides, we have now added findings from recent studies related to the topic of this study, see lines 74-82 and lines 98-108.

In the Discussion chapter:

  1. The discussion is based on a solid summary of the main findings and their comparison with previous research, but I recommend a more precise and detailed connection of the results to the existing literature. Currently, comparisons with other research are often general, which reduces the power of interpretation.  It is necessary to include a critical interpretation of the results, especially when they differ from the findings of previous studies. Consider possible reasons for such discrepancies – e.g. differences in sample, methodology, cultural context, etc. If your research results are consistent with previous research, it is important to clearly highlight this consistency and explain its implications for the scientific community and clinical practice.
  • Thank you for this comment, we have now addressed this comment further and enriched our discussion based on the reviewer’s comment, see pages 13-15.
  1. The contribution of this study to existing knowledge needs to be highlighted more clearly. Authors should indicate how their research fills gaps in the scientific literature or offers new insights into the relationship between dental anxiety, general anxiety, dental neglect and oral health-related quality of life.
  • Thank you for this comment, we have now addressed this comment further and highlighted this point in the study strengths section at the end of the discussion, see lines 373-381.
  1. The practical implications of this study are not sufficiently developed. It would be useful to highlight how the results can be applied in clinical practice, education or public health interventions.
  • Thank you for this comment, we have now addressed this comment further, see lines 387-398.
  1. I recommend that the final part of the discussion include specific suggestions for future research that logically follow from the results of this study.
  • Thank you for this comment, we have now addressed this comment further, see lines 386-389.

Round 2

Reviewer 1 Report

Comments and Suggestions for Authors

This is a MUCH stronger paper. I applaud the authors for their responsiveness. The structural equation model is both a stronger approach and an easier one for readers to absorb the multiplicity of findings at once. 

Suggestion:

The authors write, "Targeted psychosocial interventions such as cognitive-behaviour therapy are recommended in order to decrease anxiety and ultimately enhance OHRQoL."

There should a citation here. Suggestions:

Öst LG, Kvale G. Effects of cognitive behavioural treatments: A systematic review and meta-analysis. In: Öst LG, Skaret E, eds. Cognitive Behaviour Therapy for Dental Phobia and Anxiety. John Wiley & Sons; 2013:163-182.

Kvale G, Berggren U, Milgrom P. Dental fear in adults: a meta-analysis of behavioral interventions. Community Dent Oral Epidemiol. 2004;32:250-264.

Author Response

This is a MUCH stronger paper. I applaud the authors for their responsiveness. The structural equation model is both a stronger approach and an easier one for readers to absorb the multiplicity of findings at once. 

Suggestion:

The authors write, "Targeted psychosocial interventions such as cognitive-behaviour therapy are recommended in order to decrease anxiety and ultimately enhance OHRQoL."

There should a citation here. Suggestions:

Öst LG, Kvale G. Effects of cognitive behavioural treatments: A systematic review and meta-analysis. In: Öst LG, Skaret E, eds. Cognitive Behaviour Therapy for Dental Phobia and Anxiety. John Wiley & Sons; 2013:163-182.

Kvale G, Berggren U, Milgrom P. Dental fear in adults: a meta-analysis of behavioral interventions. Community Dent Oral Epidemiol. 2004;32:250-264.

Thank you, we have now cited these two important references.

Reviewer 3 Report

Comments and Suggestions for Authors

The authors have adequately addressed all the reviewers' comments.

Author Response

Thank you for confirming that we have adequately addressed all the reviewers' comments.